# Protein Language Models in Directed Evolution

**Russell Maguire** [1] [*]   **Kotryna Bloznelyte** [1] [*]   **Fikayo Adepoju** [1]   **Matthew Armean-Jones** [1]   **Shafiat Dewan** [1]
**Stella Fozzard** [1]   **Akash Gupta** [2]   **Ece Ibrahimi** [1]   **Frances Patricia Jones** [1]   **Preet Lalli** [1]   **Anna Schooneveld** [1]
**Sean Thompson** [1]   **David Berman** [1]   **Will Addison** [1]   **Luca Rossoni** [1]   **Ian Taylor** [1]

## Abstract

The dominant paradigms for integrating machine-learning into protein engineering are *de novo* protein design and guided directed evolution. Guiding directed evolution requires a model of protein fitness, but most models are only evaluated *in silico* on datasets comprising few mutations. Due to the limited number of mutations in these datasets, it is unclear how well these models can guide directed evolution efforts. We demonstrate *in vitro* how zero-shot and few-shot protein language models of fitness can be used to guide two rounds of directed evolution with simulated annealing. Our few-shot simulated annealing approach recommended enzyme variants with $1.62 \times$ improved PET degradation over 72 h period, outperforming the top engineered variant from the literature, which was $1.40 \times$ fitter than wild-type. In the second round, 240 *in vitro* examples were used for training, 32 homologous sequences were used for evolutionary context and 176 variants were evaluated for improved PET degradation, achieving a hit-rate of 39 % of variants fitter than wild-type.

## 1. Introduction

Proteins are nature's catalytic and functional materials; macro-molecules increasingly programmed in the lab for societal benefit with applications in biomaterials, diagnostics and biocatalysis.

Whilst strides have been made in *de novo* protein design in recent years (Ferruz et al., 2022; Wang et al., 2022; Watson et al., 2023), directed evolution comprising iterative mutagenesis and screening remains the primary method used for the development of optimized protein variants (Arnold, 1998;

Arnold & Volkov, 1999; Turner, 2009; Wang et al., 2021). Protein sequence space is vast and epistatic, so directed evolution experiments may require upwards of 10 000 screening samples to achieve desired protein function (Romero & Arnold, 2009; Sarkisyan et al., 2016; Hartman & Tullman-Ercek, 2019), which is a screening burden sufficient to dissuade adoption of protein engineering in smaller scale labs.

Machine-learning-guided directed evolution has emerged as a paradigm for improving the sample efficiency of protein engineering efforts (Yang et al., 2019; Biswas et al., 2021; Saito et al., 2021; Wittmann et al., 2021), and comprises two *in silico* components: a model of protein fitness and a sampling method. The majority of protein fitness models are only evaluated *in silico* on mutation effect datasets (Hopf et al., 2017; Notin et al., 2023), comprising of mostly single-mutants and with insufficient variants to compare sampling methods.

We describe how MSA Transformer (Rao et al., 2021) can be used as a few-shot model of protein fitness to sample variants using simulated annealing for machine-learning-guided directed evolution. To demonstrate this method *in vitro*, we engineered cutinases to improve polyethylene terephthalate (PET) degradation—a task of interest for protein engineering efforts with industrial relevance (Jayasekara et al., 2023; Sui et al., 2023). As controls, we compared the performance of our variants to wild-type leaf-compost cutinase (LCC) (Sulaiman et al., 2012a) and ICCM, a high-performing variant engineered through site-directed mutagenesis (Tournier et al., 2020).

We show why *in vitro* training examples are beneficial when optimizing industrially relevant phenotypes, by comparing few-shot recommended variants to zero-shot recommended variants used for training (Meier et al., 2021), and comparing both of these machine-learning-guided approaches to a global random mutagenesis baseline using error-prone polymerase chain reaction (epPCR).

---

[*]Equal contribution  [1]Cambridge Consultants, Cambridge, United Kingdom [2]University of Cambridge, Cambridge, United Kingdom.   Correspondence to:   Russell Maguire <russell.maguire@cambridgeconsultants.com>.

*Accepted at the 1st Machine Learning for Life and Material Sciences Workshop at ICML 2024.* Copyright 2024 by the author(s).

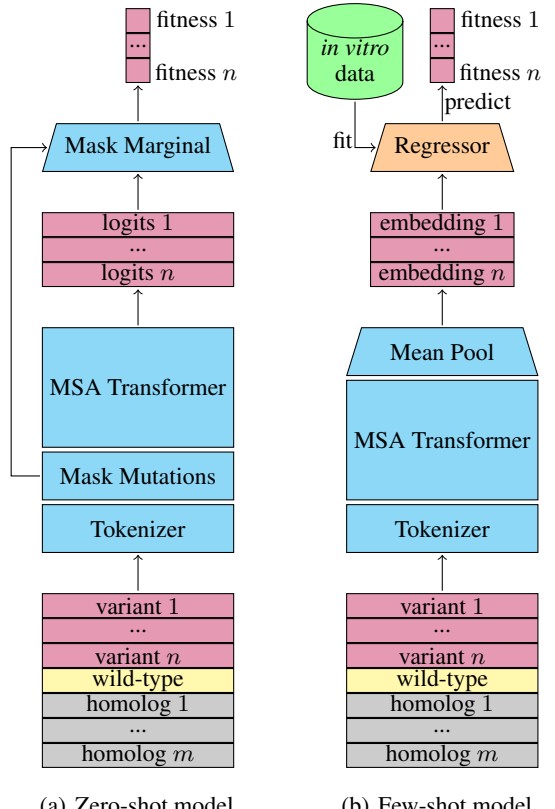

(a) Zero-shot model     (b) Few-shot model

Figure 1. MSA Transformer models of protein fitness used in machine-learning-guided directed evolution: (a) zero-shot model, where logits in mutated positions predicted are scaled relative to wild-type using the marginal objective described by Meier et al. (2021); (b) few-shot model, where the MSA Transformer weights are frozen and a small dataset of measurements are fitted to the final embeddings of MSA Transformer, using ridge regression.

## 2. Methods

### 2.1. Protein Fitness Models

In machine learning, protein language models have gained traction as tools for predicting the structure (Jumper et al., 2021; Baek et al., 2021; Lin et al., 2023) and function (Alley et al., 2019; Rao et al., 2019; Meier et al., 2021; Notin et al., 2022) of protein sequences. Protein language models are trained on public datasets of reference sequences (Suzek et al., 2015) to predict next or masked amino acids, learning the distribution of amino acids in context: UniRep (Alley et al., 2019) is a single-sequence recurrent neural network; ESM1b (Rives et al., 2021) is a single-sequence transformer encoder; and MSA Transformer (Rao et al., 2021) is a multiple-sequence transformer encoder trained to predict masked amino acids in multiple sequence alignments built from UniRef50 reference and homologous sequences.

For the prediction of function, protein language models have

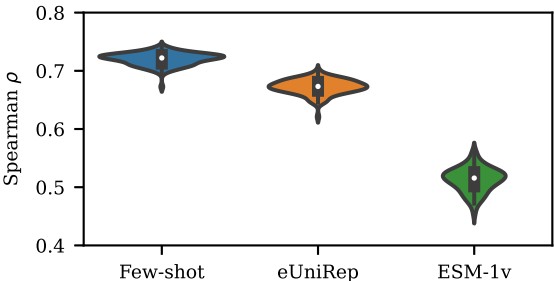

Figure 2. Few-shot MSA Transformer compared to fine-tuned UniRep (Alley et al., 2019) and non-fine-tuned ESM-1v (Meier et al., 2021) trained on samples of 240 training examples from Sarkisyan et al. (2016).

been mostly applied to zero-shot fitness prediction (Meier et al., 2021; Notin et al., 2022), where the pseudo-likelihood of mutations conditioned on the wild-type or homologous sequences are used as a fitness metric. Note, these models are not conditioned on substrates, environmental conditions, or the protein function of interest, which can include diverse characteristics from thermostability (Li et al., 2023) to substrate specificity (Goldman et al., 2022; Kroll et al., 2023).

We adapted MSA Transformer to predict the fitness of protein variants in a few-shot scenario using *in vitro* measurements. This approach is illustrated in contrast to zero-shot fitness prediction in Figure 1.

### 2.2. Multiple Sequence Alignments

MSA Transformer is unique in using explicit evolutionary context, introduced via multiple sequence alignments (MSAs), rather than relying on model weights to handle evolutionary context. This explicit evolutionary context was found to improve performance compared to single sequence in zero-shot protein fitness prediction (Meier et al., 2021). Whilst single-sequence models rely on fine-tuning on 1000s of homologous sequences to improve evolutionary context (Biswas et al., 2021), by careful selection of homologous sequences we could achieve similar predictive performance on a multiple-mutant dataset of protein variants (Sarkisyan et al., 2016) with just 32 homologs. Results are shown in Figure 2.

Following Notin et al. (2022), we identified LCC homologs using the *evcouplings* framework (Hopf et al., 2019), maximizing the number of significant evolutionary couplings with respect to the bitscore threshold of the *jackhmmer* homology search and multiple sequence alignment algorithm (Johnson et al., 2010). We sampled sequences from the resulting set of 384 homologs using sequence reweighted sampling (Hopf et al., 2017) to provide context in addi-

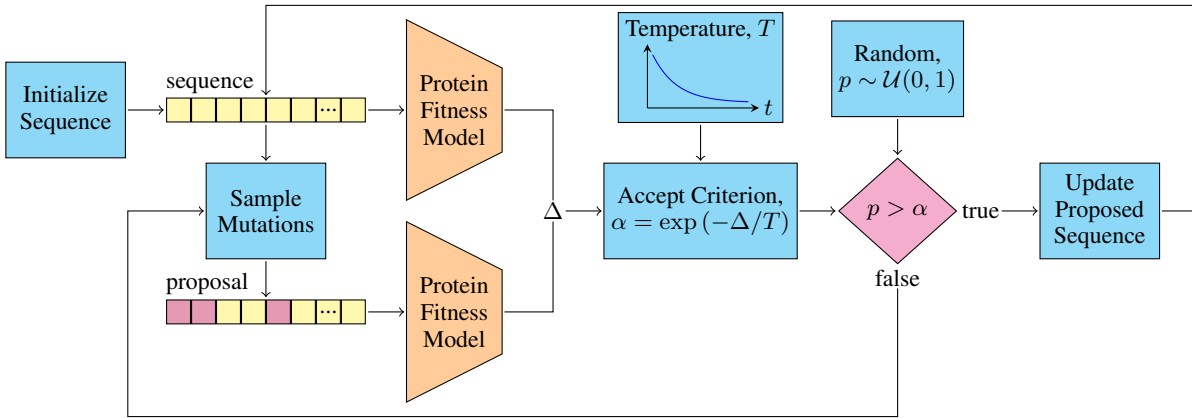

*Figure 3.* Simulated annealing used to sample high fitness variants from a model of protein fitness. This process is run until a convergence criterion is met, or a maximum number of iterations is reached.

tion to wild-type LCC and the variant sequence for MSA Transformer.

### 2.3. *In-Silco* Mutagenesis

For sampling protein variants, a common approach is to select a design window of mutations and exhaustively screen all combinations *in silico* (Saito et al., 2021; Frisby & Langmead, 2023) but this limits the diversity and number of mutations that can be explored. Generative models including autoencoders (Singer et al., 2022) and masked language models (Yamaguchi & Saito, 2022) have been used to generate a library of *de novo* variants for *in silico* screening. Biswas et al. (2021) used a simulated annealing method to propose random mutations and sample high-fitness variants using a model of protein fitness. By introducing an exponentially decaying temperature parameter, this algorithm avoids local optima close to the starting sequence and converges on high-fitness variants as the search progresses.

The simulated annealing approach used in our experiments is shown in Figure 3. Following Biswas et al. (2021), the number of mutations per iteration was sampled from a Poisson distribution with a mean from 1.0 to 1.5 mutations, the location and choice of amino acid substitution were sampled uniformly. The maximum Hamming distance from the starting sequence was constrained to a predefined trust radius. In Biswas et al. (2021), the proposed sequence rapidly approaches the trust radius and iterations are wasted until wild-type amino acids are randomly sampled for substitution. We modified the mutation sampler to first introduce the minimum number of wild-type amino acids in mutated locations to guarantee that the proposed sequences remain within the trust radius. Simulated annealing was run multiple times for 2000 iterations each, and the fittest sequence from each run was recorded and ranked. The top-k variants were selected for *in vitro* validation.

### 2.4. Experimental Setup

LCC is a wild-type cutinase used in the literature for engineering enzymes for PET degradation (Sui et al., 2023). To generate samples for lab testing, enzyme-encoding gene variants were inserted into plasmids and expressed in *E. coli* cells. Cell pellets were chemically lysed and the resulting lysates were assayed for PET degradation (1) and thermostability (2). A panel of PET degrading enzymes reported in the literature was tested under these assay conditions in-house, identifying ICCM as the best-performing variant. Thus, further experiments included wild-type LCC and ICCM as controls.

To assess PET degradation, lysate was incubated with semicrystalline PET powder and hydrolysis products TPA and MHET were quantified. Few-shot regression models were trained to predict the logarithm of the detected products relative to wild-type

$$f_{PET}(x) = \frac{c_{TPA}(x) + c_{MHET}(x)}{c_{TPA}(x_{LCC}) + c_{MHET}(x_{LCC})} \quad (1)$$

where $c_{TPA}$ and $c_{MHET}$ are the concentrations (mmol mL$^{-1}$) of TPA and MHET, respectively.

To assess variant thermostability, enzymatic activity was measured by incubating lysate with pNPB, a commonly used proxy for monitoring degradation of PET-like material (Furukawa et al., 2019). The hydrolysis product was detected chromatically and the reaction rate was calculated. The heat-treated activity was measured by pre-incubating lysate at 70 °C for 2 h, and then performing the pNPB enzymatic activity assay. Thermostability was calculated as the ratio of heat-treated to non-heat-treated activity. Few-shot regression models were trained to predict the logarithm of this ratio relative to wild-type

$$f_T(x) = \frac{\Delta A_h(x)/\Delta A_0(x)}{\Delta A_h(x_{LCC})/\Delta A_0(x_{LCC})} \quad (2)$$

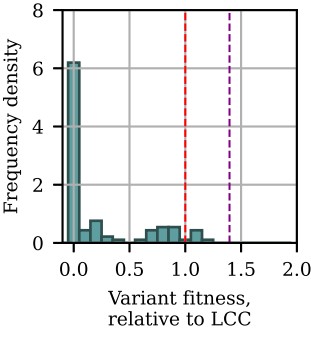

(a) Fitness of epPCR variants

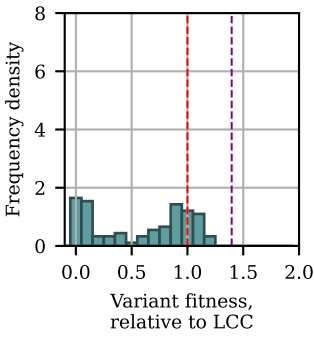

(b) Fitness of zero-shot variants

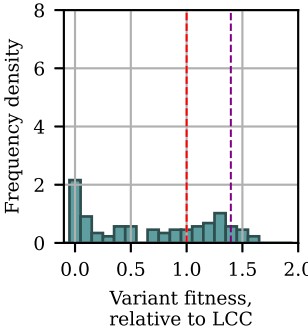

(c) Fitness of few-shot variants

*Figure 4.* Distribution of PET degredation, $f_{PET}$, for variants assayed with control LCC annotated in **red** and ICCM in **violet**. Few-shot variants are optimized for $f_{PET}$.

where $\Delta A_h$ and $\Delta A_0$ are the maximum changes in absorption at 405 nm for heat-treated and non-heat-treated samples, respectively.

LCC variants were generated using (a) epPCR, to serve as a baseline method for comparison with ML-based methods; (b) MSA Transformer zero-shot simulated annealing, to generate training data and starting point variants for few-shot approach and to serve as a comparison; and (c) MSA Transformer few-shot simulated annealing, using phenotype data collected on variants from (b). Few-shot models were trained on phenotypes (1) & (2), generating two sets of variants, one for each objective.

## 3. Results

To facilitate comparison, one 96-well plate of variants was re-assayed for PET degradation for each of the epPCR, zero-shot, and few-shot experiments, with LCC and ICCM as controls. Detailed results of these comparisons are recorded in Table 1 and Figure 4 shows the distribution of variant fitness relative to LCC. Using a binomial test, we determined that zero-shot variants were fitter than LCC more often than epPCR variants at a significance level of 0.003, and few-shot variants were fitter than LCC more often than zero-shot variants at a significance level of 0.005.

Figure 5 shows the distribution of mutations in our few-shot simulated annealing variants compared to controls. Finally, a comparison of the *in vitro* thermostability and PET degradation of variants recommended by simulated annealing is shown in Figure 6.

## 4. Discussion

Site-saturation mutagenesis is a powerful tool for exploring a small number of mutations in a focused region, but machine-learning-guided mutagenesis can explore a larger

*Table 1.* Comparison of PET degradation, $f_{PET}$, of variants from each experiment relative to wild-type LCC; hit-rate, which is the proportion of variants fitter than controls; and Spearman rank correlation, which measures repeatability across two bio-replicates. Few-shot variants are optimized for $f_{PET}$.

| | | epPCR | Zero-shot | Few-shot |
|---|---|---|---|---|
| Fitness, | mean | 0.24 | 0.58 | **0.68** |
| $f_{PET}$ | median | 0.03 | 0.71 | **0.72** |
| | max | 1.24 | 1.20 | **1.62** |
| Hit-rate, | $f_{PET} >$ LCC | 0.05 | 0.20 | **0.39** |
| | $f_{PET} >$ ICCM | 0.00 | 0.00 | **0.10** |
| Spearman, $\rho$ | | 0.89 | 0.83 | **0.94** |
| Controls, | $f_{PET}$ LCC | — | — | **1.00** |
| | $f_{PET}$ ICCM | — | — | **1.40** |

number of mutations across the scaffold and active site, and identify more beneficial mutations. In Tournier et al. (2020), ICCM was identified by recombination of four out of five beneficial mutations around the active site identified by site-saturation mutagenesis and an additional stabilizing disulfide bond. However, with only five beneficial mutation sites, there is little scope to improve ICCM further without another round of mutagenesis. In contrast, our machine-learning-guided approach identified a variant with 16 % greater catalytic activity than ICCM exploring a similar number of variants. The resulting library of beneficial mutations across the scaffold and active site could be recombined to further improve performance.

*In vitro* measurements are necessary to ensure alignment between machine-learning-guided variants and desired phenotypes. The zero-shot variants measured in our assay had a significantly higher hit-rate relative to LCC than epPCR, but variants with performance greater than ICCM were only identified after training few-shot models on those zero-shot

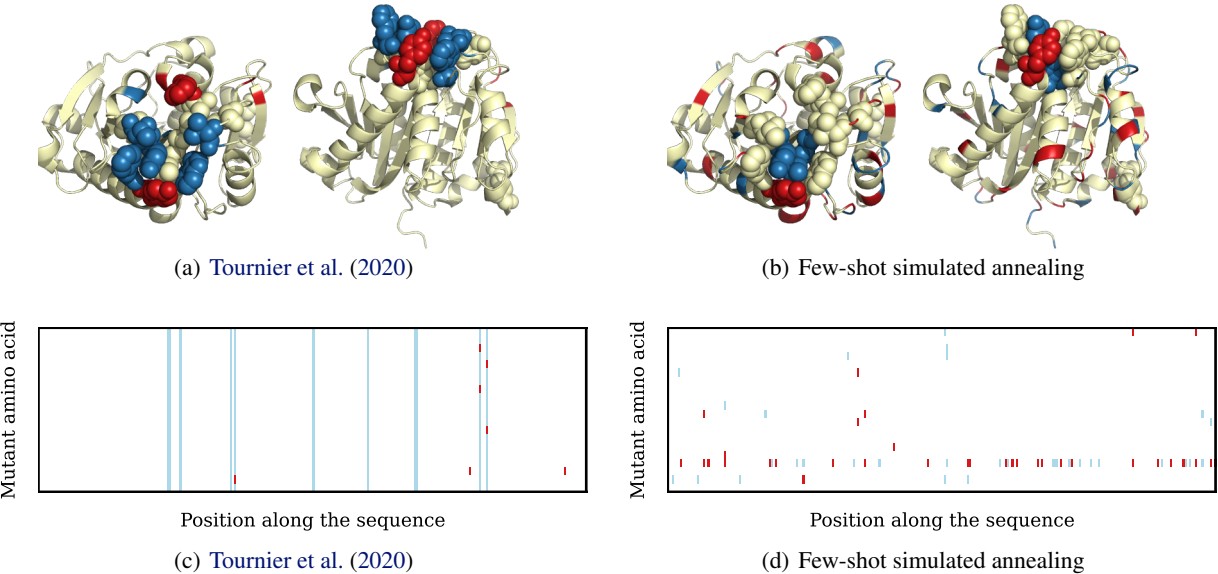

(a) Tournier et al. (2020)                    (b) Few-shot simulated annealing

(c) Tournier et al. (2020)                    (d) Few-shot simulated annealing

*Figure 5.* (a) and (b) show the distributions of mutated positions explored by few-shot simulated annealing and site saturation mutagenesis (Tournier et al., 2020) highlighted on the crystal structure of LCC obtained from the protein databank, PDB ID: 4EB0 (Sulaiman et al., 2012b). Mutations found in variants fitter than wild-type are shown in **red**, and mutations found in variants less fit than wild-type are shown in **blue**. (c) and (d) show the same mutations along the primary sequence (x-axis), but also the amino acid substituted (y-axis).

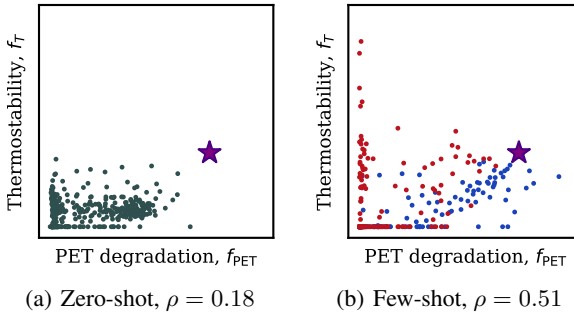

(a) Zero-shot, $\rho = 0.18$          (b) Few-shot, $\rho = 0.51$

*Figure 6.* Spearman rank correlation between *in vitro* thermostability and PET degradation of variants, ICCM annotated in **violet**. Few-shot variants optmized for thermostability are highlighted in **red** and those optimized for PET degredation are in **blue**. Variants optimized for thermostability are frequently inactive on PET.

results. One interpretation of zero-shot marginal fitness proposed by (Meier et al., 2021) is that these metrics represent the divergence between a mutated sequence and the distribution of sequences contained in the UniProt reference clusters, from which homologous sequences are identified and the underlying protein language model is trained. The majority of sequences in the UniProt knowledgebase are naturally occurring, so it is surprising that zero-shot fitness is a useful proxy for PET degradation, which performs optimally at temperatures that cannot be tolerated by most organisms. Whilst zero-shot models provide a useful screen

for early rounds of sequence exploration, there is no difference in method between zero-shot-guided mutagenesis for generating variants with improved thermostability or PET degradation, and measurements of these phenotypes were weakly correlated. In contrast, the few-shot models proposed leverage generalized features of naturally occurring protein sequences and evolutionary context, and so can be tuned to contrasting phenotypes or environmental conditions unique to industrial applications.

There is potential to extend this approach to more efficiently exploit larger libraries of fit variants identified through machine-learning-guided mutagenesis. In Tournier et al. (2020) and Biswas et al. (2021), a design window is used to restrict the explored mutation sites. We succeeded in making recommendations with a high hit-rate using a global mutagenesis method, allowing the simulated annealing algorithm to identify which locations to mutate. However, the parameter with the largest impact on hit-rate is the trust radius constraint on the maximum number of mutations introduced in each round of *in vitro* experiments, which simulated annealing cannot identify automatically. We found four mutations to be optimal for few-shot models and two for the zero-shot model. This is a significant improvement over experimental mutagenesis, where single or double mutants are screened and selectively recombined to identify variants distant from wild-type.

With further improvements, the trust radius constraint could be eliminated by introducing penalties or constraints to en-

courage the sampling of variants representative of those measured *in vitro*. For example, by modifying sampling methods to interpolate between training examples; or adopting kernel methods to model protein fitness based on similarity to high-fitness variants. With these modifications, protein language models in directed evolution could be used to effectively perform *in silico* recombination mutagenesis, identifying and combining beneficial mutations and motifs from previous lab efforts.

## Acknowledgements

We thank the reviewers for their feedback on the submitted version of the manuscript. We thank Dan McGreal, Jay Sood, Gouse Subhan Saheb and Frank Long for their expert insight managing compute infrastructure. We thank Andy Brown and Ben Johnson for helping ensure the smooth running of this project. Finally, we thank Chris Roberts, Frances Metcalfe, Richard Snell, Sally Epstein and those at Capgemini Invent for sponsoring and supporting this project and the subsequent publishing effort.

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

# A. Modeling Details

## A.1. Zero-shot Simulated Annealing Mutagenesis

To generate variants by zero-shot simulated annealing, 20 sets of 150 simulated annealing runs were performed using MSA Transformer with the masked marginal fitness objective. Simulated annealing runs were initialized with wild-type LCC, executed for 2000 iterations, and constrained to a trust radius of 2 mutations. For each iteration, the number of mutations was sampled from a Poisson distribution. The mean of the Poisson distribution was sampled uniformly from 1.0 to 1.5 mutations at the beginning of each run. Each set of 150 simulated annealing runs were executed in parallel on a different GPU and used a different sequence reweighted random sample of 32 homologs to ensure model diversity. The top variant from each run was ranked, and the top-364 variants were selected for *in vitro* validation.

Four 96 well plates were prepared with samples, each plate comprising 91 variants and 5 controls including wild-type LCC and ICCM. Two different assays were used to characterize each variant and control: PET degradation (B.3) and enzyme thermostability (B.4).

## A.2. Few-shot Simulated Annealing Mutagenesis

To generate variants by few-shot simulated annealing, 24 sets of 306 simulated annealing runs were performed using MSA Transformer with a ridge regression model, trained on results from A.1. The 24 sets comprised a different combination of parameters comprising: trust radius constraint of 2, 4, or 6 mutations; regression target (1) or (2); and initial sequence, which was of wild-type LCC, 1st, 2nd, or 3rd top variants measured in A.1. Simulated annealing was run for 2000 iterations. For each iteration, the number of mutations was sampled from a Poisson distribution. The mean of the Poisson distribution was sampled uniformly from 1.0 to 1.5 mutations at the beginning of each run. Each set of 306 simulated annealing runs was executed in parallel on a different GPU and used a different sequence reweighted random sample of 32 homologs and 240 training examples from A.1 to ensure model diversity. The top variant from each run was ranked by regression target, and the top-176 variants were selected for improved PET degradation and enzyme thermostability.

Four 96 well plates were prepared with samples, each plate comprising 88 variants and 8 controls including wild-type LCC and ICCM. Two different assays were used to characterize each variant and control: PET degradation (B.3) and enzyme thermostability (B.4).

# B. Lab Methods

## B.1. Synthesis of Variants and Expression Strains

**epPCR Variants** Error-prone polymerase chain reaction (epPCR) was used to generate libraries of LCC variants with a range of substitution frequencies. The LCC DNA sequence was first codon-optimised for expression in *E. coli*, and then synthesized and cloned into pET21(+) plasmid by Twist Bioscience.

Using pET21(+) containing LCC as the template, epPCR was then performed using a commercial kit (Agilent GeneMorph II EZClone Domain Mutagenesis Kit, 200552), adjusting conditions and parameters to achieve various mutation rates within the LCC sequence.

Plasmids containing the epPCR-generated variants were then transformed into *E. coli* XL gold cells (Agilent, 200315) and the resulting growth colonies were verified via Sanger sequencing. Plasmids were finally extracted, purified and used for the construction of expression strains. 186 Only 150 variants showing 1 to 4 amino acid substitutions were taken forward. 74 variants with 5 or more amino acid substitutions, premature stop codons, insertions or deletions were not used.

**Controls and Simulated Annealing Variants** Amino acid sequences for both control variants and machine-learning-guided variants were first codon-optimized for expression in *E. coli*, and then synthesized and cloned into pET21(+) plasmid by Twist Bioscience. Plasmids were used for the construction of expression strains.

**Construction of Expression Strains** pET21(+) plasmids containing control, machine-learning guided, and epPCR-generated variants were transformed into expression strain *E. coli* BL21 (DE3) (Sigma Aldrich). Strains were grown and stored for further testing in long-term glycerol stocks. Sanger sequencing was done on random samples to confirm the rate of successful cloning and transformation.

## B.2. Protein Expression and Sample Preparation

**Protein Expression** Seed cultures were inoculated from long-term glycerol stocks in 96 deep well plates, and grown for 18 h in 0.5 mL LB Miller with $100\,\mu g\,mL^{-1}$ carbenicillin. Seed cultures were then used to inoculate the protein expression cultures, starting OD 600 nm of 0.2. Protein expression cultures were grown in 48 deep well plates in 1.5 mL auto-induction medium (Studier, 2005) with $100\,\mu g\,mL^{-1}$ carbenicillin for 24 h. All cultures were grown at 37 °C under shaking at 800 rpm with 3 mm throw.

**Preparation of Lysate** After growth, 1.2 mL culture was transferred to a 96 deep well plate and centrifuged at

4000 xg for 20 min at 4 °C. The supernatant was removed and discarded and the pellets were frozen at −20 °C. Pellets were chemically lysed using lysis buffer consisting 300 μL B-PER Protein Extraction Reagent (Thermo Scientific) and 3 μL DNase I (Sigma). Pellets were resuspended in the lysis buffer and left shaking at 800 rpm, 3 mm throw, room temperature for 1 h. The lysates were then assayed for PET degradation and enzyme thermostability.

### B.3. PET Degradation Assay

A final volume of 1 mL lysate diluted 1:5 with Tris HCl, pH 8 was mixed with 10 mg semi-crystalline PET powder (Goodfellows) in 96 deep well plates. Reactions were incubated at 60 °C for a total of 96 h. Sampling was performed at 24 h, 48 h, 72 h and 96 h intervals. At each interval, samples were quenched with methanol containing 0.1 % v/v formic acid.

PET hydrolysis products terephthalic acid (TPA), mono(2-hydroxyethyl) terephthalate (MHET), and bis(2-hydroxyethyl) terephthalate (BHET) were then quantified by Ultra High Pressure Liquid Chromatography using an Agilent 1290 Infinity II system. Samples were filtered using Multiscreen HTS GV Filter Plates, 0.22 μm before analysis. 2 μL of filtrates were injected onto an InfinityLab Poroshell 120 EC-C18 guard column and analytical column, $2.1 \times 50$ mm, 1.9 μm (Agilent) maintained at a column compartment temperature of $(40.0 \pm 0.8)$ °C. The flow rate during analysis was 1 mL min$^{-1}$ and the mobile phase consisted of 0.1 % v/v formic acid (Phase A) and acetonitrile (Phase B). The gradient elution employed a non-linear profile: Phase A started at 6 % for 1.2 min, transitioned 6 % to 7 % for 0.4 min, then 7 % to 40 % for 1.2 min, with 40 % maintained for a further 0.2 min, before a rapid shift from 40 % to 6 % in 1 s and 6 % then held for a further 39 s.

Products were detected using a 1290 Infinity II Diode Array Detector (G7117B Detector) at a wavelength 254 nm with a peak width exceeding 0.05 min (5 Hz), 100 mAU margin for negative absorbance, and a slit width of 4 nm. Retention times for TPA, MHET and BHET peaks were 0.932 min, 1.739 min and 1.866 min, respectively. Data analysis was performed with Open Lab CDS software, using calibration curves prepared using authentic standards of each compound.

TPA and MHET concentrations at 72 h were used as a regression target for the models of protein fitness, as they were the values with the lowest experimental error. BHET concentrations were omitted as they were negligible.

### B.4. Enzyme Thermostability Assay

100 μL of lysate was incubated with 100 μL p-nitrophenyl butyrate (pNPB) into a 96 deep well plate. Hydrolysis of pNPB was followed at 405 nm for 19.5 min, sampling every 90 s using an Infinite 200 Pro plate reader (Tecan). To test enzyme thermostability, 100 μL of lysate from the same preparation was heat-treated by incubating at 70 °C for 2 h in a 96 well PCR plate with a PCR plate seal. After heat-treatment lysates were incubated with pNPB and hydrolysis followed as above. The hydrolysis reaction rates from both non-heat-treated and heat-treated samples were calculated using the maximum centered difference of 3 consecutive absorption measurements.

### B.5. Lab Automation

Transformation of *E. coli* cells, inoculation of growth plates, preparation of lysates, preparation of PET degradation assay plates, and the thermostability assay were automated on a Tecan Fluent 1080 Liquid Handling Robot.

## C. *In-Vitro* Model Performance

The results comparing the predicted fitness of variants to assay results using the model described in Figure 1 are included in Figure 7. Whilst there is no correlation between predicted fitness and assay results, these sets of variants only contain samples predicted to be significantly fitter than wild-type. The results in Table 1 suggest these models are still effective at discriminating between fit and unfit variants, sampling variants fitter than wild-type with an effective hit-rate.

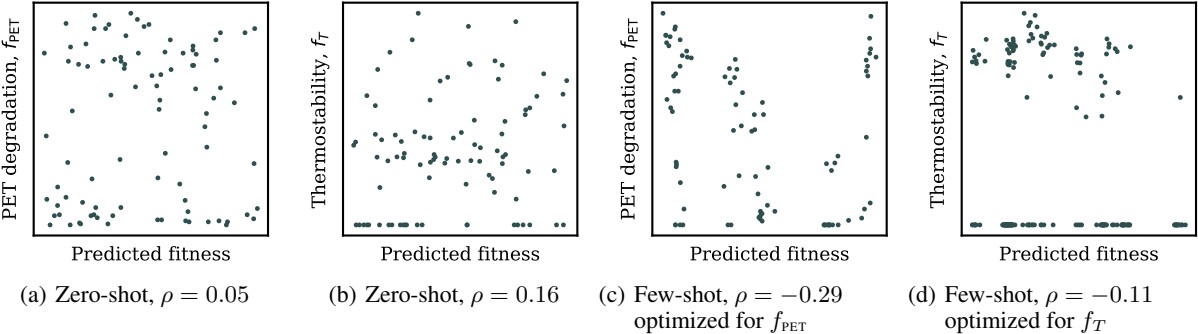

(a) Zero-shot, $\rho = 0.05$      (b) Zero-shot, $\rho = 0.16$      (c) Few-shot, $\rho = -0.29$ optimized for $f_{\mathrm{PET}}$      (d) Few-shot, $\rho = -0.11$ optimized for $f_T$

*Figure 7.* Comparison of predicted fitness to assay results for variants sampled by simulated annealing.