# OpenReview forum: "Protein Language Models in Directed Evolution"
_ICML.cc/2024/Workshop/ML4LMS — ML4LMS Poster_

### Official Review · Reviewer_97qD · 2024-06-05
**Protein Language Models in Directed Evolution**

**Rating:** 6
**Confidence:** 4

**Review:**

The authors have demonstrated that the transformer models of zero-shot and few-shot can be used effectively in sample variants with enhanced desirable qualities, such as higher catalytic activity and thermostability. The outcome was tested in vitro and they proved the effectiveness of the method. However, the predictions have huge error rate in identification of mutants.

---

### Official Review · Reviewer_oZ8m · 2024-06-10
**The paper presents a novel combination of pLMs and sampling to facilitate engineering of PET degradation enzyme with significant better performance than previous methods. Recommend acceptance.**

**Rating:** 7
**Confidence:** 3

**Review:**

# Summary:

The paper investigates the use of zero-shot and few-shot learning approaches with protein language models to guide the directed evolution of proteins. It uses MSA Transformer model to predict protein fitness and employs these predictions to enhance the degradation of PET (polyethylene terephthalate). The results demonstrate a significant improvement in PET degradation compared to traditional methods, showcasing the potential of pLMs in accelerating protein engineering. Main contributions are a novel framework of MSA Transformer-involved fitness prediction, an introduction of in vitro data into prediction pipeline, and a solid experimental validation of synthetic variants.

# Strengths and weaknesses:

**Originality**: Although the idea of combining pLMs and directed evolution is not particularly novel[1,2], this work presents a novel combination of methods - in vitro data-assisted prediction of fitness and simulated annealing sampling from fitness model.

**Quality**: This work is exceptional in its completeness - training a predictive model, sampling from the model, and experimental validation to showcase performance.

**Clarity**: The paper is well-written and easy to follow. Figures and tables are well-formatted.

**Significance**: This paper deals with an important scenario in directed evolution - few-shot optimization, which does not have satisfying solutions yet. The novel combination of pLMs and in vitro data-assisted training provides a unique perspective to tackle with this kind of problem. Overall I think this paper is well-suited for ML4LMS workshop.

**Questions**: The predictive power of MSA Transformer heavily replies on the depth of MSAs. Does this method only apply to proteins with rich resources from sequence bank? This prerequisite, along with the need for in vitro fitness data, could significantly limit the method's usability.

*References*:

[1] Damiano Sgarbossa, Umberto Lupo, Anne-Florence Bitbol (2023) Generative power of a protein language model trained on multiple sequence alignments eLife 12:e79854.

[2] Hie, B.L., Shanker, V.R., Xu, D. et al. Efficient evolution of human antibodies from general protein language models. Nat Biotechnol 42, 275–283 (2024).

---

### Official Review · Reviewer_JEQW · 2024-06-11
**An impactful in vitro evaluation of the use of zero-shot and few-shot LMs for directed evolution**

**Rating:** 8
**Confidence:** 4

**Review:**

The paper provides an impactful evaluation of the use of a pretrained protein language model for zero-shot and few-shot fitness prediction in directed evolution. The work provides compelling evidence for using a few-shot LM for fitness prediction. Still, it could benefit from clarification around the few-shot training method and performance, especially the multi-objective nature of the task.

1. Table 1 provides the results for the optimization of enzymatic activity, are there corresponding results for thermostability optimization?
2. What is the correlation of the Zero-shot and Few-shot models' prediction of activity and thermostability with the in vitro measures of the suggested variants?
3. Is there any explanation for why the variants from the Few-shot model have a greater thermostability-activity correlation than the Zero-shot model (Figure 6b) if the two properties were separately optimized for?
4. It would be nice to investigate the joint optimisation of thermostability and PET degradation.
5. Throughout, a clearer explanation is needed of the different sets of variants (optimized for activity and thermostability) - e.g. are the results in Table 1 only optimised for activity? Which of the variants in Figure 6b. are optimized for activity vs thermostabiliy?
6. Please could the authors provide more detail on the few-shot model training procedure.